# PEROV-H3: Evaluating Generative Models under Size and Symmetry Shifts in Hydrogen-Storage Perovskites

## Abstract

We introduce PEROV-H3, a rigorous benchmark targeting $ABH_3$ perovskites, designed to evaluate generative models under controlled size and symmetry shifts with structure-aware metrics. In materials science, models often excel on ideal, periodic crystals yet degrade on finite nanoparticles where size, surfaces, and edges dominate. PEROV-H3 closes this gap by pairing two complementary tasks: $(i)$ unit-cell $\rightarrow$ nanoparticle generation, probing surface- and size-dependent distortions; and $(ii)$ nanoparticle $\rightarrow$ unit-cell reconstruction, recovering bulk lattice parameters and symmetry. The benchmark comprises 100 DFT-relaxed $ABH_3$ compositions and 210,000 nanoparticle configurations spanning radii $R \in \{6, \ldots, 30\}$ Angstrom (systematic size splits for ID/OOD). Baselines reveal substantial errors under extrapolation, especially in symmetry and lattice recovery, indicating that current models memorize templates rather than learn the physics of scale. PEROV-H3 thus provides a chemically diverse, size-systematic, and structurally clean testbed for stress-testing generative models beyond bulk crystals. The dataset and the implementation are available at https://anonymous.4open.science/r/PEROV-H3.

## 1 Introduction

Materials modelling has historically been divided between two complementary but distinct regimes. The first is the regime of perfect crystals, where structures are represented by a unit cell defined through lattice constants, atomic positions, and space group symmetry (Tarantino et al., 2017). This abstraction, codified in crystallographic information files (CIFs), captures the periodic order of an infinite solid and underpins the foundations of solid-state physics and computational chemistry (Kittel & McEuen, 2018; Ashcroft & Mermin, 1976). The second regime is that of nanoparticles, finite clusters of atoms that break translational invariance. In nanoparticles, surfaces, edges, and under-coordinated sites dominate, leading to reconstructions, distortions, and quantum confinement effects that strongly alter material properties (Pizzagalli et al., 2001; Bera et al., 2010). Real materials often bridge these two representations, and understanding the transition between them is critical for predicting catalytic activity, optical response, stability, and electronic behavior (Vergara et al., 2017).

Despite advances in computational modelling and machine learning, bridging these regimes remains a major challenge (Li et al., 2023). Models trained exclusively on bulk data typically perform well at reproducing unit cell properties, but degrade when asked to generate nanoparticles or to reconstruct unit cells from nanoparticle inputs (Gleason et al., 2024). Errors include misidentification of space group symmetry, inaccurate lattice parameters, and an inability to capture size-dependent surface reconstructions. Existing benchmarks reinforce these limitations: CSPBench, for example, has shown that even state-of-the-art crystal structure prediction algorithms frequently fail unless test cases are closely aligned with their training distributions (Wei et al., 2024). Other datasets in the perovskite domain emphasize targeted applications such as band gap prediction or photovoltaic efficiency, but do not evaluate bidirectional structure conversion or systematic size variation (Pollice et al., 2021; Kusaba et al., 2022).

To address this gap, we propose *PEROV-H3*, an evaluation framework explicitly designed to test both nanoparticle generation from unit cells and unit cell reconstruction from nanoparticles. The framework consists of 100 chemically diverse $ABH_3$ perovskite compounds—a family of materials that has been extensively investigated for hydrogen storage applications (Kuo et al., 2024; Ahsin et al.,

2020). Each compound is represented by a bulk unit cell and a large collection of systematically generated nanoparticles spanning radii from 6 Å to 30 Å. By focusing on a consistent perovskite motif, the framework allows controlled variation in size and chemistry without introducing confounding structural classes. Two tasks define the evaluation. The forward task challenges models to generate nanoparticles of specified size from a given unit cell, requiring accurate treatment of surface relaxation and finite-size distortions. The inverse task requires reconstructing the unit cell from a nanoparticle, testing whether models can identify underlying symmetry and lattice constants despite surface noise. Crucially, PEROV-H3 employs in-distribution splits for intermediate sizes and out-of-distribution splits for extreme sizes, providing a clear test of extrapolative generalization. Baseline experiments using state-of-the-art methods demonstrate that while existing approaches perform adequately at intermediate radii, their accuracy deteriorates at small and large extremes, with failures most pronounced in symmetry recovery and lattice parameter prediction (Chen & Ong, 2021; Li et al., 2021). By coupling systematic nanoparticle variation with clean, paired unit cell representations, PEROV-H3 provides a rigorous and chemically diverse benchmark for assessing whether models can genuinely learn the physics of scale rather than interpolate between familiar cases.

## 2 RELATED WORK

### 2.1 CRYSTALS: UNIT CELLS AND NANOPARTICLES

Crystalline solids are defined by their unit cell, the smallest repeating motif specified by lattice constants, space group symmetry, and atomic basis (Anosova et al., 2024). This description underpins crystallography, density functional theory (DFT), and the majority of large materials databases (Carbogno et al., 2022; Hellenbrandt, 2004). The unit cell abstraction is powerful because it condenses infinite periodic order into a compact blueprint (Jain et al., 2013). However, real materials rarely manifest exclusively as perfect crystals (Baig et al., 2021). At the nanoscale, finite clusters of atoms form nanoparticles, where translational symmetry is broken and surfaces dominate (Cheng et al., 2024; Ye et al., 2024). In these systems, under-coordinated atoms, surface reconstructions, and edge distortions significantly alter structural and functional behavior (Zhang et al., 2023). Understanding the correspondence between unit cells and nanoparticles is essential for many applications (Cheng et al., 2024). Catalysis, for example, depends on surface terminations and defects, while optical properties in perovskite quantum dots depend on quantum confinement (Ye et al., 2024). Yet modelling nanoparticles from bulk inputs—or recovering bulk parameters from nanoparticle structures—remains a fundamental challenge due to the scale gap and the nonlinear effects introduced by surfaces (Zhang et al., 2023).

### 2.2 DATASETS FOR STRUCTURAL MODELLING

Several large-scale datasets have driven progress in computational materials science. The Materials Project (Jain et al., 2013) and the OQMD (Kirklin et al., 2015) provide millions of crystal structures in CIF format, supporting both supervised learning of material properties and unsupervised exploration of chemical space. CSPBench (Wei et al., 2024) offers curated benchmarks for crystal structure prediction, focusing on stability ranking and bulk structure recovery. PubChemQC (Kim et al., 2025) extends quantum chemical calculations to millions of molecules, enabling cross-domain learning. Perov-5 (Castelli et al., 2012a;b) provides a large-scale collection of perovskite structures and properties, while OC20 (Chanussot et al., 2021) connects relaxed metal-surface structures with atomic forces and OC22 (Tran et al., 2023) further expands surface reaction benchmarks. CrysMTM (Polat et al., 2025) introduces a multitask benchmark focused on crystal graph representations. While these resources have been transformative, they remain limited to bulk representations and do not include nanoparticles or systematic size variation. Other benchmarks have expanded to test robustness and generalization. Matbench (Dunn et al., 2020) defines a suite of supervised property prediction tasks spanning multiple datasets. In the perovskite domain, specialized datasets have been constructed for photovoltaic efficiency, thermodynamic stability, and band-gap prediction. These collections support important application-driven tasks but are not designed to probe the structural transition from bulk to nanoparticles. At present, no dataset pairs each unit cell with systematically generated nanoparticles across a controlled range of radii. This gap limits the evaluation of models that aim to learn the physics of scale or to transfer knowledge between bulk and finite systems. The lack of such benchmarks motivates the design of PEROV-H3.

## 2.3 Models Leveraging Existing Datasets

Machine learning methods in materials science have advanced rapidly through access to bulk crystal datasets. Early work focused on graph neural networks such as CGCNN (Xie & Grossman, 2018) and message passing neural networks (Klipfel et al., 2023). More recently, equivariant neural networks such as DimeNet (Gasteiger et al., 2020), PaiNN (Schütt et al., 2021), and NequIP (Batzner et al., 2022) have achieved state-of-the-art performance on force and energy prediction tasks by enforcing rotational and translational symmetries. Generative approaches, including variational autoencoders (Luo et al., 2024), diffusion models (Khastagir et al., 2025), LLM based flow models Sriram et al. (2024), and autoregressive graph models (Antunes et al., 2024), have been proposed for crystal structure generation and inverse design. Without datasets that link unit cells and nanoparticles, it is impossible to rigorously evaluate how well models generalize across size and scale. PEROV-H3 addresses this by providing paired data and well-defined tasks for nanoparticle generation from unit cells and unit cell reconstruction from nanoparticles, thereby filling a critical gap in the benchmarking ecosystem.

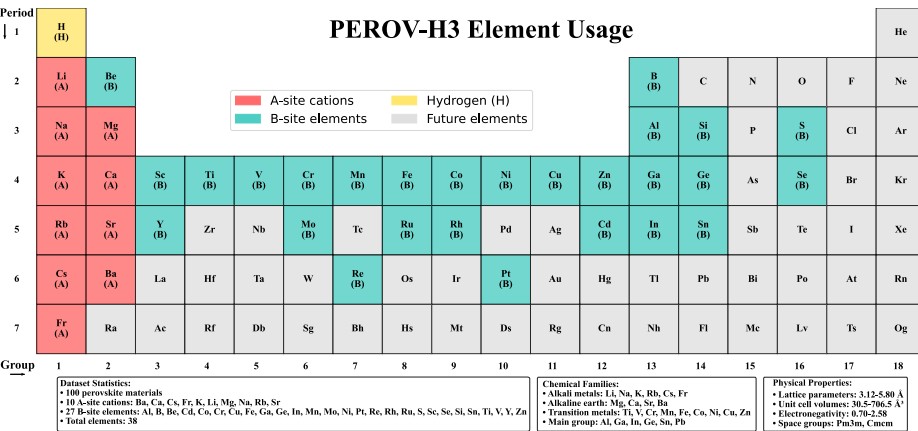

Figure 1: Comprehensive element usage analysis of the PEROV-H3 dataset for hydrogen storage applications. The periodic table visualization highlights the systematic chemical diversity across 100 ABH$_3$ perovskite materials, where A-site cations (blue) include 10 alkali and alkaline earth metals, B-site elements (red) encompass 27 transition metals and main group elements, and hydrogen (green) serves as the X-site anion. The dataset represents two distinct space groups (Pm3m and Cmcm), reflecting both cubic and orthorhombic perovskite structures. Physical properties span lattice parameters from 3.12 to 5.80 Å, unit cell volumes from 30.5 to 706.5 Å$^3$, and electronegativity values from 0.70 to 2.58, demonstrating the broad compositional and structural space covered. This systematic exploration enables comprehensive evaluation of structure-property relationships in perovskite hydrogen storage materials, supporting generative model development for both unit cell to nanoparticle structure prediction and nanoparticle to unit cell reconstruction tasks.

## 3 Framework Creation

### 3.1 Nanoparticle construction

We curated 100 distinct ABH$_3$ compositions with high hydrogen-storage potential and retrieved their *DFT-relaxed* lattice parameters from the literature and open databases. From the corresponding CIFs we first built the primitive unit cell, then expanded to a supercell that accommodates the target radius. Finite nanoparticles were obtained by retaining atoms within a sphere of radius $R$ centered at $x_0$. No additional relaxation or surface/ligand modeling was applied, thereby isolating size/geometry effects while remaining consistent with vetted bulk parameters.

**Supercell size.** For each ABH$_3$ composition, we constructed a $20 \times 20 \times 20$ supercell by replicating the primitive unit cell along $a, b, c$, yielding box lengths $L_i = 20\, a_i$ (i$\in \{a, b, c\}$). This satisfies $L_i \geq 2R_{\max} + \Delta$ for the maximum carving radius $R_{\max} = 30\,$Å with a safety margin $\Delta \approx 5$–$10\,$Å, ensuring carved nanoparticles remain well-separated from periodic images.

**Spherical carving.** A finite nanoparticle of target radius $R$ is carved by retaining all atoms whose Cartesian positions fall inside a sphere of radius $R$ centered at a chosen origin,

$$R \in \{6, 7, 8, \ldots, 30\}\,\text{Å}, \qquad \mathcal{C}_R = \{\, \mathbf{x}_i \in \mathbb{R}^3 \mid \|\mathbf{x}_i - \mathbf{x}_0\| \leq R \,\}.$$

This yields a controlled family of clusters per composition, covering the transition from strongly surface-dominated (small $R$) to near-bulk behavior (large $R$). Elements included in PEROV-H3 are shared in Figure 1.

### 3.2 ROTATION SAMPLING AND DATASET SPLITS

To remove orientation bias and enforce strict separation between training and evaluation, each nanoparticle is augmented by rigid rotations sampled on SO(3) (Shoemake, 1985). Rotations are represented by unit quaternions $q \in \mathbb{H}$, $\|q\| = 1$. The geodesic angle between two rotations $q_i, q_j$ is

$$d(q_i, q_j) = 2\arccos\big(|\langle q_i, q_j \rangle|\big),$$

where $\langle \cdot, \cdot \rangle$ is the Euclidean inner product in $\mathbb{R}^4$. A greedy sampler generates a set $\mathcal{Q}(\theta)$ such that

$$d(q_i, q_j) \geq \theta \quad \forall\, i \neq j,$$

with angular spacing $\theta$ controlling density. Approximating coverage by spherical caps on SO(3) gives the heuristic

$$N(\theta) \approx \frac{4\pi}{2\pi\,(1 - \cos\theta)} = \frac{2}{1 - \cos\theta},$$

so that coarser spacings produce fewer orientations and denser spacings produce more (Kuffner, 2004). For the spacings used here,

$$N(15°) \approx 59, \qquad N(12°) \approx 92, \qquad N(9°) \approx 163.$$

Let $\mathcal{Q}_{\text{train}}$ denote the fixed training grid (seeded deterministically). To guarantee disjointness, candidate evaluation quaternions are accepted only if they satisfy an exclusion margin from the training set,

$$d(q, q') \geq \delta_{\text{split}} \quad \forall\, q' \in \mathcal{Q}_{\text{train}},$$

with $\delta_{\text{ID}} = 6°$ and $\delta_{\text{OOD}} = 4.5°$. Because $d(q_i, q_j) = 2\arccos(|\langle q_i, q_j \rangle|)$, these constraints are equivalent to

$$|\langle q, q' \rangle| \leq \cos\big(\tfrac{1}{2}\delta_{\text{split}}\big).$$

Fixed left-multiplications $R_{\text{ID}}$ and $R_{\text{OOD}}$ are applied to all ID and OOD rotations, respectively,

$$q_{\text{eff}} = R_{\text{split}} \cdot q,$$

using deterministic offsets (e.g., Euler $(6°, 8°, 12°)$ for ID and $(15°, 25°, 35°)$ for OOD) to further decorrelate orientations without affecting geodesic distances.

Radius-based partitions are

$$\mathcal{R}_{\text{ID}} = \{10, 11, 17, 21, 24, 26\}, \qquad \mathcal{R}_{\text{OOD}} = \{6, 7, 29, 30\},$$

with all remaining radii used for training. Training uses $\theta_{\text{train}} = 15°$, ID uses $\theta_{\text{ID}} = 12°$, and OOD uses $\theta_{\text{OOD}} = 9°$.

### 3.3 DESIGN RATIONALE FOR SPLIT VALUES AND ANGULAR SPACINGS

The choice of radii and angular spacings follows a coarse-to-dense principle that balances learnability, statistical power, and fairness. Training spans the interior of the radius range with a coarse orientation lattice ($\theta_{\text{train}} = 15°$), while ID tests use mid-range radii $(10, 11, 17, 21, 24, 26)$ and a denser lattice ($\theta_{\text{ID}} = 12°$, about 92 orientations) to stabilize metrics without adding diversity. The OOD set probes extremes $(6, 7, 29, 30)$, where errors are harder due to scaling ($S/V \sim 3/R$), and uses an even denser lattice ($\theta_{\text{OOD}} = 9°$, about 163 orientations) to reduce estimator variance. In all cases, evaluation orientations are kept at nonzero geodesic distance from training via $\delta_{\text{ID}}$ and $\delta_{\text{OOD}}$, guaranteeing no overlap. Formally, for an error functional $E$ and rotation operator $\mathcal{R}_q$, orientation-averaged performance is

$$\overline{E}(R) = \frac{1}{|\mathcal{Q}_R|} \sum_{q \in \mathcal{Q}_R} E(\mathcal{R}_q(\text{prediction}),\ \mathcal{R}_q(\text{reference})),$$

and the denser grids reduce variance of $\overline{E}(R)$. Highly symmetric structures are further deduplicated by retaining only one representative quaternion per unique rotation. Full split details are in Appendix A.1.

### 3.4 Chemical Composition Analysis

Figure 2 summarizes the chemical composition of PEROV-H3 across A- and B-site sublattices. The A-site contains ten elements, mainly alkali and alkaline earth metals, with Li (21), Na (16), and K (14) most common. Their electronegativity distribution has a mean of $0.999 \pm 0.218$, while ionic radii are broadly dispersed ($1.160 \pm 0.300$ Å), reflecting size-driven variability that strongly impacts distortions and stability. The B-site is more chemically diverse, spanning 27 species dominated by transition metals and metalloids, led by V and Rh (8 each), followed by Cu and Zn (6 each). Unlike the A-site, its properties are more uniform, with electronegativity $1.543 \pm 0.114$ and radii $0.694 \pm 0.040$ Å, indicating chemical variety without extreme size mismatches. In total, the dataset covers 85 distinct A–B combinations, most frequently Li–V, offering a balanced space for probing both size- and chemistry-driven effects.

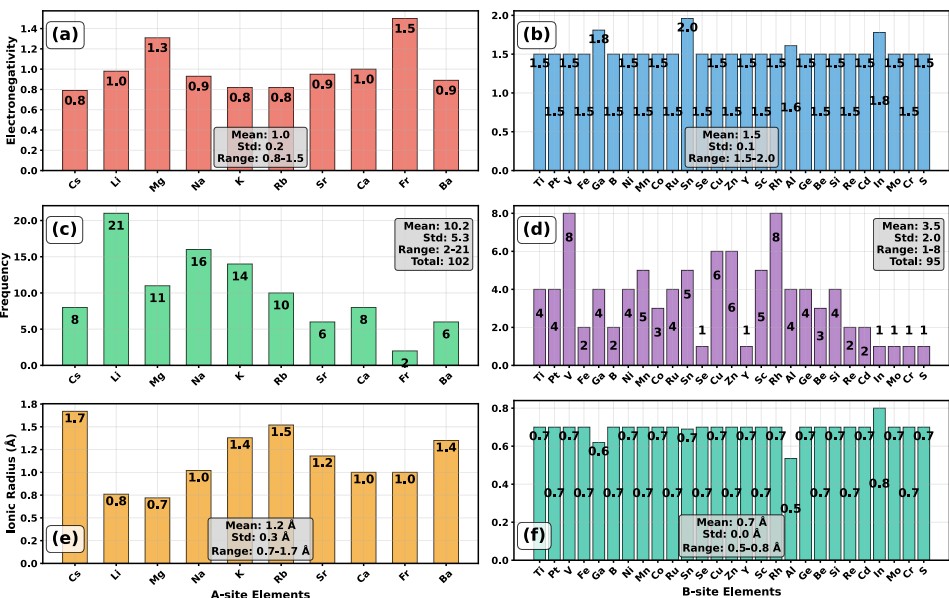

Figure 2: Chemical composition analysis of PEROV-H3 materials showing element distributions, electronegativity, and ionic radius properties. (a) A-site element electronegativity distribution (n=10 elements) with mean 0.999 and standard deviation 0.218, dominated by alkali and alkaline earth metals. (b) B-site element electronegativity distribution (n=27 elements) with mean 1.543 and standard deviation 0.114, showing transition metals and metalloids. (c) A-site element frequency distribution with Li being the most common (21 occurrences), followed by Na (16) and K (14). (d) B-site element frequency distribution with V and Rh being most common (8 occurrences each), followed by Cu and Zn (6 each). (e) A-site ionic radius distribution with mean 1.160 Å and standard deviation 0.300 Å, reflecting the size diversity of A-site cations. (f) B-site ionic radius distribution with mean 0.694 Å and standard deviation 0.040 Å, showing more uniform sizes for B-site elements. The dataset contains 85 unique A-B combinations from 100 materials, with Li-V being the most frequent combination. Bar values are positioned alternately for optimal readability.

### 3.5 Size Analysis

Figure 3 shows the scaling of PEROV-H3 nanoparticles across 25 radii ($R = 6.0$–$30.0$ Å), where atom counts grow from 81 to 10,408 (mean $3333 \pm 3118$) and volumes follow $V = \frac{4}{3}\pi R^3$, ranging from 491.7 to 107,498.7 Å$^3$ (mean $33,520.9 \pm 32,299.7$). Surface areas scale quadratically as $S = 4\pi R^2$, spanning 325.7–11,007.0 Å$^2$ (mean 4501.9). Structural ratios reveal strong size effects: surface-to-volume decreases from 0.668 Å$^{-1}$ at $R = 6$ to 0.102 Å$^{-1}$ at $R = 30$ (mean 0.230), density remains bulk-like at $9.060 \times 10^{-24}$ g/Å$^3$ with low variance, and per-atom volumes average 9.323 Å$^3$/atom (bounded 6.092–10.328). These results confirm that PEROV-H3 adheres closely to analytic scaling laws while preserving realistic statistical variation, providing a rigorous platform for testing nanoparticle models across length scales.

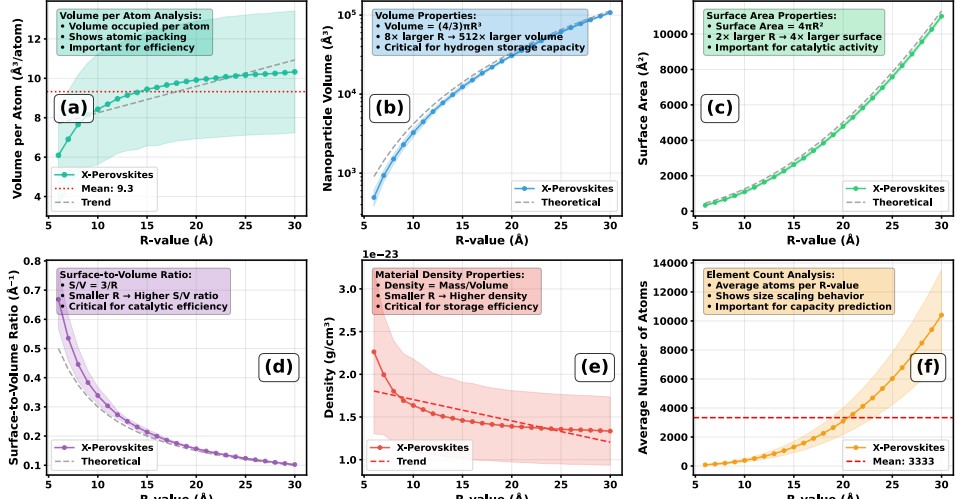

Figure 3: Size analysis of PEROV-H3 materials showing the relationship between R-value and various structural properties. (a) Volume per atom analysis revealing atomic packing efficiency and density variations across different R-values. The dataset comprises 25 R-values ranging from 6.0 to 30.0 Å, with corresponding volumes from 491.7 to 107,498.7 Å$^3$ and surface areas from 325.7 to 11,007.0 Å$^2$. (b) Volume scaling behavior following the theoretical V = (4/3)πR$^3$ relationship, with experimental data showing excellent agreement. (c) Surface area distribution following the S = 4πR$^2$ scaling law, critical for catalytic activity assessment. (d) Surface-to-volume ratio (S/V = 3/R) distribution, indicating higher efficiency for smaller R-values. (e) Material density distribution as a function of R-value, demonstrating the inverse cubic relationship with R-value. (f) Atom count distribution showing the scaling relationship between R-value and total atomic content. Error bars represent standard deviations across multiple samples.

## 3.6 CRYSTALLOGRAPHIC STRUCTURE ANALYSIS

Figure 4 summarizes lattice parameter distributions and correlations for the 100 unit cells in PEROV-H3. The $a$ and $c$ parameters are narrowly distributed (means 3.858 Å and 3.882 Å, std. $< 0.6$ Å), while $b$ varies widely from 3.124–17.230 Å (std. 1.844 Å), reflecting flexibility in certain chemistries. Unit cell volumes span 30.480–706.533 Å$^3$ (mean 69.046 Å$^3$), capturing both compact and expanded perovskite phases. Correlation analysis shows $a$ and $c$ are most strongly coupled ($r = 0.967$), consistent with cubic/pseudocubic symmetry, followed by $b$–$c$ ($r = 0.887$) and $a$–$b$ ($r = 0.741$), with distortions along $b$ driving orthorhombicity. These results highlight that PEROV-H3 balances near-cubic metrics with chemically induced distortions, offering a structurally rich landscape to evaluate algorithms for lattice prediction and reconstruction.

## 3.7 TASK DEFINITIONS

The evaluation is organized around two complementary transformations: one moving from the crystallographic unit cell to a finite nanoparticle, and the other reversing that process. Both directions are essential for testing whether models capture the physics that governs scale transitions. Performance is assessed on both in-distribution radii (interpolation) and out-of-distribution radii (extrapolation).

**Task 1: From Unit Cell to Nanoparticle.** The forward mapping begins with a primitive unit cell, denoted $\mathcal{U}_m$, together with a radius parameter $R$. The goal is to construct a particle $\mathcal{P}$ in three-dimensional space that exhibits both the periodic ordering encoded in $\mathcal{U}_m$ and the finite-size surface distortions induced by truncation at radius $R$. Formally,

$$\mathcal{F}_1 : (\mathcal{U}_m, R) \longrightarrow \mathcal{P} \subset \mathbb{R}^3. \tag{1}$$

Accuracy of $\mathcal{F}_1$ is quantified by comparing the predicted particle $\mathcal{P}$ with the reference structure $\mathcal{P}^\dagger$ using geometric and structural measures:

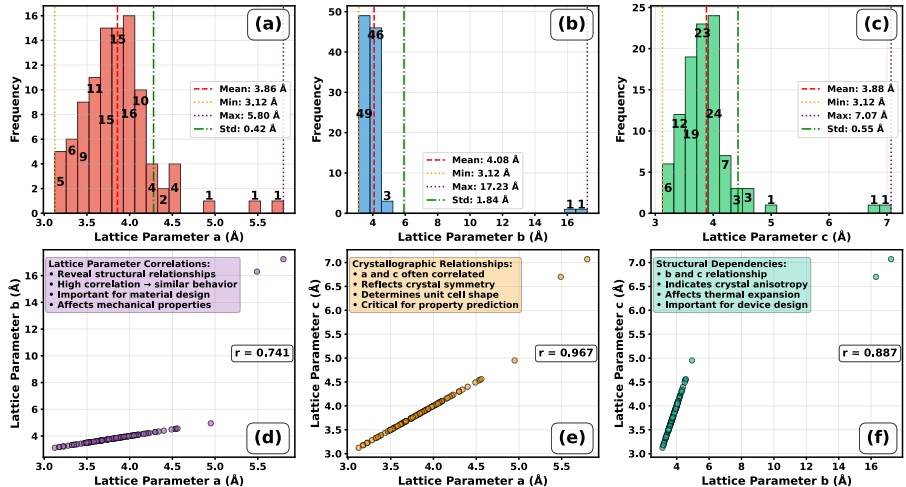

Figure 4: Crystallographic structure analysis of PEROV-H3 materials showing lattice parameter distributions and correlations. (a) Lattice parameter a distribution (n=100 samples) with mean 3.858 Å and range 3.124-5.800 Å. (b) Lattice parameter b distribution with mean 4.080 Å and range 3.124-17.230 Å, showing the largest variation. (c) Lattice parameter c distribution with mean 3.882 Åand range 3.124-7.070 Å. (d) Correlation between lattice parameters a and b (r = 0.741), revealing structural relationships important for material design. (e) Strong correlation between lattice parameters a and c (r = 0.967), indicating crystallographic symmetry constraints. (f) Correlation between lattice parameters b and c (r = 0.887), showing structural dependencies affecting thermal expansion properties. Statistical lines indicate mean (red dashed), min/max (orange/purple dotted), and standard deviation (green dash-dot). The alternating bar value positioning ensures optimal readability across the parameter ranges.

**Root-mean-square deviation (RMSD).**

$$\text{RMSD}(\mathcal{P}, \mathcal{P}^\dagger) = \sqrt{\frac{1}{N} \sum_{i=1}^{N} \|\mathbf{r}_i - \mathbf{r}_i^\dagger\|^2}, \tag{2}$$

where $N$ denotes the number of atoms, $\mathbf{r}_i \in \mathcal{P}$ and $\mathbf{r}_i^\dagger \in \mathcal{P}^\dagger$ represent the atomic coordinates after optimal alignment. This measures coordinate-level similarity.

**Hausdorff distance.**

$$d_{\text{Haus}}(\mathcal{P}, \mathcal{P}^\dagger) = \max \left\{ \sup_{\mathbf{p} \in \mathcal{P}} \inf_{\mathbf{q} \in \mathcal{P}^\dagger} \|\mathbf{p} - \mathbf{q}\|, \ \sup_{\mathbf{q} \in \mathcal{P}^\dagger} \inf_{\mathbf{p} \in \mathcal{P}} \|\mathbf{q} - \mathbf{p}\| \right\}, \tag{3}$$

where $\mathbf{p}$ and $\mathbf{q}$ represent points on the particle surfaces. This quantifies the worst-case geometric discrepancy.

**Convex hull volume difference.**

$$\Delta V_{\text{hull}}(\mathcal{P}, \mathcal{P}^\dagger) = \frac{\left| V\big(\text{Hull}(\mathcal{P})\big) - V\big(\text{Hull}(\mathcal{P}^\dagger)\big) \right|}{V\big(\text{Hull}(\mathcal{P}^\dagger)\big)}, \tag{4}$$

where $\text{Hull}(\cdot)$ is the convex hull of a particle and $V(\cdot)$ its volume. This captures global shape fidelity.

**Radial distribution function error.**

$$E_{\text{RDF}}(\mathcal{P}, \mathcal{P}^\dagger) = \int_0^{R_{\text{max}}} \big(g_{\mathcal{P}}(r) - g_{\mathcal{P}^\dagger}(r)\big)^2 \, dr, \tag{5}$$

where $g_{\mathcal{P}}(r)$ and $g_{\mathcal{P}^\dagger}(r)$ are the radial distribution functions of $\mathcal{P}$ and $\mathcal{P}^\dagger$, and $R_{\text{max}}$ is a cutoff radius. This assesses differences in local atomic environments.

**Local environment variance.**

$$\text{V}_{\text{R}}(\mathcal{P}) = \frac{\sigma^2\big(\{v_i : v_i \in \mathcal{N}_R(\mathbf{r}_i)\}\big)}{\mu^2\big(\{v_i : v_i \in \mathcal{N}_R(\mathbf{r}_i)\}\big)}, \tag{6}$$

where $\mathcal{N}_R(\mathbf{r}_i)$ defines the set of atoms within radius $R$ of site $\mathbf{r}_i$, $v_i$ represents the coordination number of $\mathbf{r}_i$, $\sigma^2(\cdot)$ stands for the variance, and $\mu(\cdot)$ denotes the mean. This measures heterogeneity of local environments.

Together, these metrics capture coordinate-level similarity, global shape fidelity, and consistency of local atomic arrangements.

**Task 2: From Nanoparticle to Lattice.** The inverse transformation seeks to recover crystallographic invariants from a finite particle. Given a nanoparticle $\mathcal{P}$, the model must infer both the lattice parameters and the symmetry group that define the underlying periodic structure. In compact form,

$$\mathcal{F}_2 : \mathcal{P} \longrightarrow \big(\Lambda = (a, b, c, \alpha, \beta, \gamma),\ \Gamma \in \mathcal{S}\big), \tag{7}$$

where $\Lambda$ denotes the six lattice constants (edge lengths $a, b, c$ and interaxial angles $\alpha, \beta, \gamma$) and $\Gamma$ is an element of the crystallographic space-group set $\mathcal{S}$.

Evaluation of $\mathcal{F}_2$ compares predictions $(\Lambda, \Gamma)$ to the ground truth $(\Lambda^\dagger, \Gamma^\dagger)$ using the following metrics:

**Root-mean-square error of lattice parameters.**

$$\mathrm{RMSE}(\Lambda, \Lambda^\dagger) = \sqrt{\frac{1}{6} \sum_{i=1}^{6} \big(\lambda_i - \lambda_i^\dagger\big)^2}, \tag{8}$$

where $\Lambda = (\lambda_1, \ldots, \lambda_6)$ are the predicted lattice constants and $\Lambda^\dagger = (\lambda_1^\dagger, \ldots, \lambda_6^\dagger)$ are the true values. This metric quantifies the average deviation in lattice geometry.

**Space-group classification accuracy.**

$$\mathbb{1}[\Gamma = \Gamma^\dagger] = \begin{cases} 1, & \Gamma = \Gamma^\dagger \\ 0, & \text{otherwise,} \end{cases} \tag{9}$$

where $\Gamma$ is the predicted space group and $\Gamma^\dagger$ the true space group. This indicator reports whether the symmetry class is correctly identified.

**Joint recovery accuracy.**

$$\mathbb{1}[(\Lambda, \Gamma) = (\Lambda^\dagger, \Gamma^\dagger)] = \begin{cases} 1, & \Lambda = \Lambda^\dagger\ \wedge\ \Gamma = \Gamma^\dagger \\ 0, & \text{otherwise.} \end{cases} \tag{10}$$

This stricter indicator requires both the lattice constants and the space group to be simultaneously correct.

Together, $\mathcal{F}_1$ and $\mathcal{F}_2$ constitute a bidirectional probe of scale transfer. The first tests whether models can enrich a minimal crystallographic blueprint into a realistic finite particle; the second examines whether local geometric signals suffice to reconstruct global periodic order. Only by succeeding in both directions can a model demonstrate true mastery of the crystal-to-nanoparticle continuum.

# 4 EXPERIMENTS

Multiple generative models— such as CDVAE (Xie et al., 2021), DiffCSP (Jiao et al., 2023), FlowMM (Miller et al., 2024), MatterGen-MP (Zeni et al., 2023), and ADiT (Joshi et al., 2025)—are evaluated on all three tasks using the splits defined in Section 3.7. PEROV-H3 has been adapted to be compatible with the PyTorch Geometric Fey & Lenssen (2019) library and the model implementations are developed using PyTorch Paszke (2019) and model's respective official libraries. The details of the implementation and the model settings are shared in Appendix B while comprehensive experimental analysis for each task are presented in Appendix C.

## 4.1 TASK 1: UNIT CELL TO NANOPARTICLE GENERATION

Table 1 in Appendix C.1 shows that most Task 1 models achieve deceptively low losses ($\sim 0.01$) but collapse on structure, with RMSD and Hausdorff errors near $40$–$85\,\text{Å}$ and hull deviations of $3 \times 10^4$–$5 \times 10^4$. By contrast, CDVAE, despite its higher nominal loss ($\sim 0.10$), reconstructs faithfully, achieving RMSD $\sim 0.006$–$0.007\,\text{Å}$, Hausdorff $\sim 0.020\,\text{Å}$, and hull errors below 10. Crucially, this advantage holds slightly under OOD evaluation, where all other models degrade catastrophically, highlighting the difficulty of PEROV-H3.

## 4.2 Task 2: Nanoparticle to Lattice Inference

As detailed in the supplementary results (Appendix C.2), all baselines fail on Task 2, which requires recovering both lattice parameters and space group from a nanoparticle. DiffCSP and ADiT reach strong in-distribution space-group accuracy (0.987 and 0.980, respectively), yet both yield extremely poor lattice predictions (RMSE $\approx 63.7\,\text{Å}$) and zero joint recovery. CDVAE performs better on lattice regression (RMSE $32.5 \pm 0.8\,\text{Å}$ for both ID and OOD) but still collapses to $0.0$ joint accuracy. FlowMM is even less stable, with RMSE $62.6 \pm 3.7\,\text{Å}$ and space-group accuracy $0.0$ across ID and OOD. MatterGen similarly plateaus at RMSE $63.7\,\text{Å}$ with $0.980$ SG accuracy but no joint success. These results highlight that, despite high classification scores, no baseline achieves correct lattice and symmetry simultaneously, and OOD performance remains indistinguishable from ID, showing that PEROV-H3 reliably exposes the inability of current methods to generalize lattice recovery under distributional shift.

## 5 Limitations

Despite its systematic design, PEROV-H3 remains an idealized framework rather than a direct representation of experimental data. All nanoparticles are carved from perfect supercells without incorporating thermal fluctuations, defects, or surface reconstructions that naturally arise during synthesis. This makes the dataset highly controlled but also less representative of the imperfections and environmental influences that shape real nanostructures. As a result, model performance measured on this framework may not fully reflect robustness under experimental conditions.

In addition, the framework is restricted to the $ABH_3$ perovskite prototype and only two space groups ($Pm\bar{3}m$ and $Cmcm$). While this structural consistency facilitates rigorous benchmarking, it narrows the diversity relative to the broader chemical and crystallographic landscape. Nanoparticle generation is further constrained by deterministic carving procedures that enforce geometric uniformity but omit ligand effects and energy-driven relaxation. The evaluation splits, based solely on particle size and rotational sampling, provide a strong test of scale generalization but do not probe other challenges such as compositional extrapolation or defect tolerance. These choices should be recognized as trade-offs: they enhance clarity and reproducibility but limit the scope of conclusions that can be drawn.

## 6 Conclusion and Future Work

PEROV-H3 establishes a systematic evaluation framework for bridging the gap between crystallographic unit cells and finite nanoparticles. Its controlled chemical diversity, broad range of radii, and structural consistency make it especially suited for studying nanoscale phenomena central to hydrogen storage. In particular, the dataset captures how particle size, surface-to-volume ratio, and lattice symmetry impact stability and surface reactivity—factors that directly influence hydrogen absorption and release kinetics. Current analyses demonstrate that models trained on PEROV-H3 can already provide reliable structural predictions relevant to hydrogen storage applications, positioning the framework as a valuable tool for accelerating the design of efficient storage materials.

Future work will build on this foundation to expand both scope and realism. Incorporating additional space groups, mixed-anion variants, and defect-engineered perovskites will broaden the coverage of hydrogen storage chemistries beyond the current $ABH_3$ family. Extending the framework with temperature-dependent structures and relaxed surfaces will further align predictions with experimental conditions, enhancing relevance for practical storage environments. Finally, integrating property-focused benchmarks—such as hydrogen binding energies, diffusion pathways, and cyclic stability—will strengthen the framework's role as both a structural evaluation tool and a driver of discovery in hydrogen storage research. In this way, PEROV-H3 not only demonstrates strong current utility but also provides a clear trajectory toward becoming a comprehensive benchmark for the next generation of hydrogen energy materials.

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
