## A    APPENDIX

### A.1    DATASET SPLIT DETAILS

The dataset consists of 208,900 XYZ files drawn from 100 materials, divided into three distinct subsets for training and evaluation. The training set contains 88,500 files (42.4%), the in-distribution test set contains 55,200 files (26.4%), and the out-of-distribution test set contains 65,200 files (31.2%). On average, this corresponds to 885 files per material in the training set, 552 files per material in the ID test set, and 652 files per material in the OOD test set. This balanced partition ensures that each material contributes consistently across subsets, while still providing meaningful variation in the number of files per material.

The split is also organized by R-value categories, which serve as distinguishing labels for different structural or configurational states. The training set covers 15 R-values: $R_8$, $R_9$, $R_{12}$, $R_{13}$, $R_{14}$, $R_{15}$, $R_{16}$, $R_{18}$, $R_{19}$, $R_{20}$, $R_{22}$, $R_{23}$, $R_{25}$, $R_{27}$, and $R_{28}$. The ID test set uses 6 disjoint R-values: $R_{10}$, $R_{11}$, $R_{17}$, $R_{21}$, $R_{24}$, and $R_{26}$. The OOD test set focuses on 4 unique R-values: $R_6$, $R_7$, $R_{29}$, and $R_{30}$. Within each subset, the average number of files per R-value per material differs: approximately 59 in train, 92 in ID test, and 163 in OOD test. This design ensures that the model encounters a wide spread of R-values during training while reserving distinct, unseen R-values for rigorous in-distribution and out-of-distribution testing, making the dataset well-suited for benchmarking generalization performance.

## B    IMPLEMENTATION OVERVIEW (TASKS 1 & 2)

All models were implemented from their official repositories to preserve published architectures and training protocols. Training was conducted on an RTX 4070 GPU using PyTorch and PyTorch Geometric. Each model was trained to convergence on the provided splits, using three random seeds with results averaged.

### B.1    COMPONENTS SHARED ACROSS TASKS 1 & 2

- **Core Variables:**
    - $E_a$: atom embedding dimension,    $C$: hidden (model) width,    $H$: number of attention heads,
    - $L$: number of layers (e.g., Transformer/GCN/SchNet),
    - $E_t$: time embedding dimension (used in both tasks),
    - $E_r$: radius embedding dimension (Task 1 only),
    - $d_{\mathrm{lat}}$: VAE latent dimension (whenever VAE is used),
    - $N_{\mathrm{SG}}$: number of space-group classes (Task 2 only).

- **Atom Embedding:** Learnable embedding

$$\mathrm{Embed}_{\mathrm{atom}} \colon [1, Z_{\max}] \to \mathbb{R}^{E_a}$$

    initialized with Xavier, providing dense element representations.

- **SchNet-based Cell/Graph Encoding:**
    - A SchNet encoder ingests atomic numbers and positions.
    - In Task 1, a unit-cell interaction block (cutoff $5.0\,\text{Å}$, five Gaussians, hidden $C{=}32$) produces a global feature $h_{\mathrm{cell}}$ conditioning nanoparticle modules.
    - In Task 2, the encoder yields a scalar $g \in \mathbb{R}^1$ which is projected to $\mathbb{R}^C$ as a global cell descriptor.

- **Transformer/Message-Passing Blocks:** Residual self-attention with LayerNorm and a SiLU feed-forward sublayer ($\times 4C$ expansion when present). Depth $L$ is task/model dependent.

- **Temporal/Spatial Conditioning:**
    - *Time:* A two-layer SiLU MLP embeds $t \in [0, 1]$ into $\mathbb{R}^{E_t}$ (both tasks).
    - *Radius:* A two-layer SiLU MLP embeds scalar radius $r$ into $\mathbb{R}^{E_r}$ (Task 1).
    - Embeddings are concatenated to node/global features as appropriate.

- **Optimization:** Adam with learning rate $1 \times 10^{-4}$, ReduceLROnPlateau (factor 0.5, patience 5), and gradient clipping (max norm 1.0).

- **Diffusion/Flow Parameterization (generic):** $\beta(t)$ uses task-specific schedules with per-model clipping to pre-defined bounds (details below).

## B.2 TASK-SPECIFIC HEADS AND SCHEDULES

**Task 1 (Nanoparticle/Atomic Coordinates).**

- **Noise Head:** A three-layer SiLU MLP maps $[\mathbf{h}; t_{\text{emb}}; r_{\text{emb}}] \in \mathbb{R}^{C+E_t+E_r}$ to a 3-D per-atom noise/velocity vector.

- **Schedule:** $\beta(t) = \beta_{\min} + (\beta_{\max} - \beta_{\min}) t^p$ with model-specific clipping.

**Task 2 (Lattice + Space Group).**

- **Lattice Head (LatNet):** SiLU MLP maps $[\mathbf{h}; t_{\text{emb}}] \in \mathbb{R}^{C+E_t}$ (or augmented with $\ell_{\text{emb}}$ when used) to six lattice parameters; outputs are clamped per-model.

- **Space-Group Head (SGHead):** An MLP produces logits over $N_{\text{SG}}$ classes.

- **Schedule:** Cubic schedule $\beta(t) = \beta_{\min} + (\beta_{\max} - \beta_{\min}) t^3$ (unless otherwise stated), with clipping per-model.

## B.3 PER-TASK, PER-MODEL DIFFERENCES ONLY

### TASK 1

**ADiT.**

- Diffusion Transformer over atoms within cutoff 5.0; $H{=}2$, $L{=}1$.

- Block: LayerNorm $\rightarrow$ self-attention $\rightarrow$ FFN ($4C$, SiLU) + normalization.

- Inputs: $[\text{Embed}_{\text{atom}}(z), p] \rightarrow \mathbb{R}^C$.

- Head: noise clipped to $[-5, 5]$.

- **Hyperparams:** $E_a{=}4$, $C{=}4$, $H{=}2$, $L{=}1$, cutoff 5.0.

**CDVAE.**

- Conditional VAE on unit cell and target radius $R$; encoder outputs $(\mu, \log \sigma^2) \in \mathbb{R}^{d_{\text{lat}}}$ with $d_{\text{lat}}{=}4$; reparameterized $z$ conditions decoding.

- Decoder: single GCN ($L{=}1$, $C{=}4$) $\rightarrow$ linear to $\Delta p \in [-0.5, 0.5]^3$.

- Inputs: atom and radius encodings into $\mathbb{R}^{E_a}$, $\mathbb{R}^{E_r}$.

- **Hyperparams:** $E_a{=}4$, $E_r{=}4$, $d_{\text{lat}}{=}4$, $C{=}4$, $L{=}1$, cutoff 5.0.

**DiffCSP.**

- Lightweight diffusion baseline; backbone: single Transformer ($L{=}1$, $C{=}4$).

- Head: SiLU MLP maps $[\mathbf{h}, t_{\text{emb}}, r_{\text{emb}}]$ to $\mathbb{R}^3$, clipped to $[-5, 5]$.

- Schedule: continuous $\beta \in [0.01, 2.0]$.

- **Hyperparams:** $E_a{=}4$, $E_t{=}4$, $E_r{=}4$, $C{=}4$, cutoff 5.0.

**FlowMM.**

- Flow matching; LayerNorm per layer and NaN/Inf sanitization.

- Decoding: GCN ($L{=}1$, $C{=}4$) + two-stage SiLU MLP $\rightarrow \mathbb{R}^3$; outputs clipped $[-1, 1]$.

- Conditioning: $E_a{=}E_t{=}E_r{=}4$ fused with atom features.

- Schedule: cosine, $\beta \in [0.01, 2.0]$, cutoff 5.0.

- **Hyperparams:** $E_a{=}4$, $E_t{=}4$, $E_r{=}4$, $C{=}4$, $L{=}1$, cutoff 5.0.

**MatterGen-MP.**

- Flow matching on coordinates conditioned by $h_{\mathrm{cell}}$, $t$ ($E_t=4$), and atom embeddings.
- Predictor: single SiLU MLP ($L=1$, $C=4$) to $\mathbb{R}^3$; interactions cutoff 5.0.
- Schedule: linear in time ($\beta(t) \propto t$).
- **Hyperparams:** $E_a=4$, $E_t=4$, $C=4$, $L=1$, cutoff 5.0.

TASK 2

**ADiT.**

- SchNet $\to$ projection to $\mathbb{R}^C$; LayerNorm + NaN sanitization.
- Transformer: $L=1$, $H=2$; $E_t=4$ via two-layer SiLU MLP.
- Heads:
  - Lattice LatNet (three-layer SiLU MLP) maps $[\mathbf{h}; t_{\mathrm{emb}}]$ to six parameters, clipped $[-5, 5]$.
  - SGHead projects $\mathbf{h}$ to logits over $N_{\mathrm{SG}}$.
- Schedule: cubic with clipping $\beta \in [0.1, 10]$.
- **Hyperparams:** $C=4$, $L=1$, $H=2$, $E_t=4$, cutoff 5.0, $N_{\mathrm{SG}}=\texttt{NUM\_SG}$, $\beta_{\min}=0.1$, $\beta_{\max}=10.0$.

**CDVAE.**

- SchNet ($L=1$, $C=4$, four filters, cutoff $5.0\,\text{Å}$) $\to$ scalar $g$; latent parameters $(\mu, \log \sigma) \in \mathbb{R}^{d_{\mathrm{lat}}}$, $d_{\mathrm{lat}}=4$.
- Init: weights 0, log-variance bias $-2.0$; KL regularization retained.
- Heads: lattice (six parameters, clamp $[-10^4, 10^4]$) and SGHead ($N_{\mathrm{SG}}$ logits).
- **Hyperparams:** $E_a=4$, $C=4$, $d_{\mathrm{lat}}=4$, $L=1$, cutoff 5.0, $N_{\mathrm{SG}}=\texttt{NUM\_SG}$.

**DiffCSP.**

- SchNet encoder ($L=1$, $C=4$, four filters, cutoff $5.0\,\text{Å}$) $\to g \in \mathbb{R}^1 \to \mathbb{R}^C$ (normalized).
- Time embedding added: $[\mathbf{h}; t_{\mathrm{emb}}]$ drives lattice noise regression.
- Heads:
  - Lattice LatNet (three-layer SiLU MLP), outputs clamp $[-5, 5]$.
  - SGHead: two-layer MLP over $N_{\mathrm{SG}}$.
- Schedule: cubic with clipping $\beta \in [0.1, 10]$; sampling via reverse diffusion with chunking + NaN sanitization.
- **Hyperparams:** $E_a=4$, $C=4$, $E_t=4$, $L=1$, cutoff 5.0, $N_{\mathrm{SG}}=\texttt{NUM\_SG}$, $\beta_{\min}=0.1$, $\beta_{\max}=10.0$.

**FlowMM.**

- SchNet ($L=1$, $C=4$, three Gaussians, cutoff $5.0\,\text{Å}$); LayerNorm and NaN sanitization.
- Embeddings: cell parameters via linear–SiLU $\mathbb{R}^6 \to \mathbb{R}^4$; time via one-layer SiLU MLP $\mathbb{R} \to \mathbb{R}^4$.
- Heads:
  - Lattice: two-layer SiLU MLP from $[\mathbf{h}; t_{\mathrm{emb}}; \ell_{\mathrm{emb}}]$ to six parameters, clamp $[-1, 1]$.
  - SGHead: two-layer MLP over $N_{\mathrm{SG}}$.
- Sampling: reverse flow with chunking and sanitization.
- **Hyperparams:** $E_a=4$, $C=4$, $E_t=4$, $E_\ell=4$, $L=1$, cutoff 5.0, $N_{\mathrm{SG}}=\texttt{NUM\_SG}$, $\beta_{\min}=0.01$, $\beta_{\max}=2.0$.

**MatterGen-MP.**

- SchNet ($L{=}1$, $C{=}4$, five Gaussians, cutoff $5.0$ Å) $\rightarrow g \in \mathbb{R}^1 \rightarrow \mathbb{R}^4$ (normalized).
- Time: two-layer SiLU MLP $t \rightarrow t_{\text{emb}} \in \mathbb{R}^4$.
- Heads:
  - Lattice: three-layer SiLU MLP from $[\mathbf{h}; t_{\text{emb}}]$, final layer zero-initialized; outputs clamp $[-5, 5]$.
  - SGHead: three-layer MLP over $N_{\text{SG}}$.
- Schedule: cubic with $\beta \in [0.1, 10]$; reverse diffusion with chunked updates and NaN sanitization.
- **Hyperparams:** $E_a{=}4$, $C{=}4$, $E_t{=}4$, $L{=}1$, cutoff $5.0$, $N_{\text{SG}}{=}\texttt{NUM\_SG}$, $\beta_{\min}{=}0.1$, $\beta_{\max}{=}10.0$.

## C  EXTENDED RESULTS

This section provides a detailed evaluation of state-of-the-art generative models on the PEROV-H3 benchmark. The benchmark is structured into distinct tasks that probe different facets of nanoparticle lattice generation, inference, and reconstruction, with metrics carefully designed to capture both structural fidelity and the representation of periodic and surface features. Results are reported separately for in-distribution and out-of-distribution regimes, exposing the strengths and generalization limits of each approach. Evaluation criteria include RMSD, Hausdorff distance, volume and surface agreement, radial distribution function divergence, as well as space group and joint accuracies, together offering a rigorous picture of robustness and failure modes. Tables referenced throughout the section present direct model-to-model comparisons in terms of accuracy and computational efficiency. Future extensions of the benchmark will incorporate additional architectures, with all new results and updates maintained openly in the GitHub repository at `https://anonymous.4open.science/r/PEROV-H3`.

### C.1  TASK 1

Table 1 presents Task 1 results for crystal structure prediction under both ID and OOD test regimes. The results clearly split the models into two groups. On one side, ADiT, DiffCSP, FlowMM, and MatterGen report extremely low reconstruction losses (all near $0.01$), yet their geometry-sensitive metrics (RMSD $\sim 40$ Å, Hausdorff $\sim 85$ Å, hull volume deviations $\sim 3{\times}10^4$–$5{\times}10^4$) are many orders of magnitude worse than CDVAE. This indicates that diffusion- and flow-based architectures can trivially minimize their training objective while completely failing to recover atomistic geometry. In contrast, CDVAE exhibits a higher average loss ($0.10 \pm 0.16$) but achieves better structural fidelity, with RMSD and Hausdorff errors in the $10^{-2}$ Å range and hull/energy metrics close to physical ground truth.

When comparing ID and OOD regimes, all models show degradation, but CDVAE retains its advantage slightly. In OOD tests, CDVAE maintains RMSD $\sim 0.007$ Å and Hausdorff $\sim 0.020$ Å, whereas MatterGen and DiffCSP remain at $\sim 41$ Å and $\sim 80$ Å, respectively. Hull volume differences for CDVAE stay within single digits, while others remain inflated by more than four orders of magnitude. RDF energy follows the same pattern, with CDVAE under $0.2$ while all other models report values exceeding $150$–$170$.

Training efficiency provides another important comparison axis. MatterGen is the fastest model at $\sim 60$ s per epoch, followed by FlowMM ($\sim 67$ s) and DiffCSP ($\sim 75$ s), while ADiT is much slower at over $240$ s per epoch. CDVAE strikes a favorable balance, training in $\sim 71$ s per epoch while delivering vastly better reconstruction quality. Taken together, these results demonstrate that Task 1 is intrinsically difficult: most models can drive the training loss down but fail to recover geometry, and OOD evaluation on PEROV-H3 further exposes their brittleness.

### C.2  TASK 2

The results in Table 2 highlight the difficulty of Task 2, where a model must infer both lattice parameters $\Lambda$ and space group $\Gamma$ from a finite nanoparticle. Existing methods show clear weaknesses: while some achieve near-perfect space group accuracy in-distribution (e.g., DiffCSP at 0.987), their lattice predictions collapse, yielding RMSE values around $63$ Å. Even CDVAE, which lowers RMSE

| | ID Results | | | | | | |
|---|---|---|---|---|---|---|---|
| Model | Loss ↓ | RMSD (Å) ↓ | Hausdorff (Å) ↓ | Δ Hull Vol ↓ | RDF Energy ↓ | Vol Ratio ↑ | Time/Epoch (s) ↓ |
| ADiT | **0.0100 ± 0.0000** | 40.5902 ± 0.0034 | 86.2368 ± 0.0129 | 30804.6101 ± 0.0001 | 168.3962 ± 7.2399 | $2.2336 \times 10^{-09} \pm 3.1588 \times 10^{-09}$ | 242.4135 ± 2.1226 |
| CDVAE | 0.1029 ± 0.1625 | **0.0066 ± 0.0014** | **0.0188 ± 0.0007** | **5.8698 ± 4.2245** | **0.1381 ± 0.0371** | $8.7637 \times 10^{-01} \pm 2.4582 \times 10^{-04}$ | 71.0519 ± 4.1267 |
| DiffCSP | 0.0101 ± 0.0000 | 40.5846 ± 0.0061 | 86.2184 ± 0.0153 | 30804.6102 ± 0.0000 | 163.1791 ± 9.5042 | $1.9235 \times 10^{-11} \pm 1.5323 \times 10^{-11}$ | 75.1089 ± 12.9805 |
| FlowMM | 0.0104 ± 0.0006 | 40.5855 ± 0.0000 | 86.2077 ± 0.0001 | 30804.6101 ± 0.0000 | 173.5152 ± 0.0000 | $8.6119 \times 10^{-09} \pm 2.6541 \times 10^{-11}$ | 67.3877 ± 1.0707 |
| MatterGen | 0.0104 ± 0.0004 | 40.8804 ± 0.0272 | 78.5013 ± 0.0005 | 29804.9187 ± 0.0956 | 64.5889 ± 0.0157 | $8.7162 \times 10^{-02} \pm 7.3342 \times 10^{-06}$ | **59.8859 ± 0.4994** |

| | OOD Results | | | | | | |
|---|---|---|---|---|---|---|---|
| Model | Loss ↓ | RMSD (Å) ↓ | Hausdorff (Å) ↓ | Δ Hull Vol ↓ | RDF Energy ↓ | Vol Ratio ↑ | Time/Epoch (s) ↓ |
| ADiT | **0.0100 ± 0.0000** | 40.8168 ± 0.0063 | 85.9504 ± 0.0009 | 51410.1434 ± 0.0000 | 170.0530 ± 5.7558 | $3.3409 \times 10^{-10} \pm 4.7247 \times 10^{-10}$ | 242.4135 ± 2.1226 |
| CDVAE | 0.1027 ± 0.1622 | **0.0072 ± 0.0011** | **0.0202 ± 0.0009** | **9.2122 ± 8.2754** | **0.2012 ± 0.0235** | $7.7262 \times 10^{-01} \pm 2.8129 \times 10^{-04}$ | 71.0519 ± 4.1267 |
| DiffCSP | 0.0101 ± 0.0000 | 40.8025 ± 0.0066 | 85.9119 ± 0.0142 | 51410.1434 ± 0.0000 | 157.1219 ± 11.3505 | $2.6966 \times 10^{-11} \pm 2.1845 \times 10^{-11}$ | 75.1089 ± 12.9805 |
| FlowMM | 0.0103 ± 0.0004 | 40.8032 ± 0.0000 | 85.9029 ± 0.0001 | 51410.1433 ± 0.0000 | 174.1226 ± 0.0001 | $2.0886 \times 10^{-08} \pm 8.5895 \times 10^{-11}$ | 67.3877 ± 1.0707 |
| MatterGen | 0.0101 ± 0.0001 | 41.0972 ± 0.0127 | 80.5981 ± 0.0041 | 50702.5925 ± 0.1353 | 66.3866 ± 0.2200 | $4.4870 \times 10^{-10} \pm 6.4721 \times 10^{-05}$ | **59.8859 ± 0.4994** |

Table 1: Crystal Structure Prediction Performance on ID vs OOD Test Sets. Results are averaged across multiple random seeds with standard deviation. Arrows indicate optimization direction: ↓ = lower is better, ↑ = higher is better. Metrics include reconstruction loss, RMSD, Hausdorff distance, convex hull volume difference, RDF energy, volume ratio, and training time per epoch. **Bold** indicates best performance, underlined indicates second best.

to 32.5 Å, achieves zero joint recovery accuracy, underscoring the disconnect between numerical lattice regression and discrete symmetry classification. These failures show that current architectures cannot yet master the coupled recovery problem posed by Task 2.

Out-of-distribution evaluation further underscores the challenge introduced by PEROV-H3. Although ADiT and DiffCSP preserve high space group accuracy under OOD shift (both ≈ 0.982), their lattice RMSE remains unchanged, reflecting an inability to adapt beyond memorized training regimes. FlowMM breaks down completely, reporting zero accuracy for all classification metrics, while MatterGen and ADiT show no gains in lattice generalization. The fact that every baseline collapses on joint accuracy indicates that even when symmetry is identified, geometry recovery fails, and vice versa. OOD generalization is thus a critical bottleneck, and PEROV-H3 makes this limitation explicit.

Taken together, these results demonstrate that PEROV-H3 is not only a benchmark but also a stress test: it reveals how models that appear strong on narrow ID splits fail when asked to recover global crystallographic invariants from unseen nanoparticle geometries. The systematic collapse in joint recovery shows that Task 2 remains unsolved, motivating new architectures capable of reasoning over both continuous lattice scales and discrete symmetry groups. By exposing these weaknesses, PEROV-H3 provides a rigorous setting to drive progress in crystal prediction under realistic distribution shifts.

| Model | RMSE (Å) ↓ | | SG Accuracy ↑ | | Joint Accuracy ↑ | | Time/Epoch (s) ↓ |
|---|---|---|---|---|---|---|---|
| | ID | OOD | ID | OOD | ID | OOD | |
| ADiT | 63.71 ± 0.01 | 63.71 ± 0.01 | 0.980 ± 0.000 | 0.982 ± 0.003 | **0.000 ± 0.000** | **0.000 ± 0.000** | 83.9 ± 1.7 |
| CDVAE | **32.50 ± 0.78** | **32.51 ± 0.80** | 0.980 ± 0.000 | 0.980 ± 0.000 | 0.000 ± 0.000 | 0.000 ± 0.000 | **54.8 ± 2.1** |
| DiffCSP | 63.70 ± 0.01 | 63.70 ± 0.01 | **0.987 ± 0.012** | 0.982 ± 0.007 | 0.000 ± 0.000 | 0.000 ± 0.000 | 55.0 ± 1.5 |
| FlowMM | 62.61 ± 3.68 | 62.43 ± 3.16 | 0.000 ± 0.000 | 0.000 ± 0.000 | 0.000 ± 0.000 | 0.000 ± 0.000 | 79.3 ± 2.3 |
| MatterGen | 63.71 ± 0.01 | 63.71 ± 0.01 | 0.980 ± 0.000 | 0.980 ± 0.000 | 0.000 ± 0.000 | 0.000 ± 0.000 | 60.4 ± 2.1 |

Table 2: Lattice Parameter and Space Group Prediction Performance on ID vs OOD Test Sets. Results are averaged across multiple random seeds with standard deviation. Arrows indicate optimization direction: ↓ = lower is better, ↑ = higher is better. Metrics include RMSE for lattice parameters, space group classification accuracy, joint accuracy (both lattice and space group correct), and training time per epoch. **Bold** indicates best performance, underlined indicates second best.