# OpenReview forum: "PEROV-H3: Evaluating Generative Models under Size and Symmetry Shifts in Hydrogen-Storage Perovskites"
_ICLR.cc/2026/Conference — ICLR 2026 Conference Withdrawn Submission_

### Official Review · Reviewer_6gjW · 2025-10-27

**Soundness:** 3
**Presentation:** 2
**Contribution:** 2
**Rating:** 2
**Confidence:** 4

**Summary:**

This paper introduces PEROV-H3, a benchmark designed to evaluate generative models under size and symmetry shifts using a family of ABH₃ perovskites relevant to hydrogen storage.
The benchmark pairs two complementary tasks:
- Unit cell → nanoparticle generation, which tests whether models can generate finite nanoparticles of various radii from a crystalline prototype;
- Nanoparticle → unit cell reconstruction, which tests whether models can infer lattice parameters and symmetry from a finite structure.

The dataset includes 100 DFT-relaxed perovskite unit cells and 210,000 geometrically truncated nanoparticles covering radii from 6–30 Å.
Several generative models (CDVAE, DiffCSP, FlowMM, MatterGen, ADiT) are evaluated, showing severe degradation under size extrapolation.
The authors claim that PEROV-H3 fills a key gap in testing whether generative models “learn the physics of scale.”

**Strengths:**

- The benchmark explicitly tests scale generalization and symmetry recovery, which are rarely considered in existing crystal-structure datasets (e.g., Matbench, CSPBench).
- The construction pipeline, rotational sampling on SO(3), and radius-based ID/OOD splits are carefully described and mathematically formalized.
- A paired dataset linking bulk and finite systems could be useful for future work on nanoscale materials modeling, surface property prediction, and hydrogen storage.

**Weaknesses:**

- The paper does not clearly explain why AI is needed for nanoparticle prediction. There is no comparison with existing physical methods or evidence that conventional simulations (e.g., DFT, Wulff construction) are too slow or limited. As a result, the motivation feels artificial rather than driven by a real computational bottleneck.
- Nanoparticles are generated by spherical truncation without surface relaxation or energy minimization, ignoring well-known physical models such as Wulff construction. This makes the dataset geometrically clean but physically unrealistic.
- The experiments occupy less than two pages and provide no quantitative tables or visual examples. All models are said to “fail,” but no diagnostic analysis or insight is provided about why they fail or how their architectures behave differently.
- Since every baseline model collapses, the benchmark does not reveal differentiating strengths or weaknesses. It is unclear whether the failure stems from task mismatch, data design, or missing inductive biases.
-The dataset includes only ABH₃ compositions and two space groups, with no energy or property labels. This restricts both scientific applicability and the potential to evaluate physically meaningful model performance.

**Questions:**

- How does the proposed benchmark address a real computational or scientific bottleneck compared to existing physical methods such as DFT-based Wulff construction or classical MD simulations?
- What is the expected benefit or acceleration from using AI models on this task—has any runtime or efficiency comparison been made?
- Since the nanoparticles are generated by geometric truncation without surface relaxation, how physically meaningful are these structures for evaluating generative models?
- All evaluated models were designed for periodic crystals—does their failure reflect flaws in the models or a mismatch with the task definition?
- Why were only two space groups (Pm-3m, Cmcm) and the ABH₃ composition family chosen? Would the benchmark generalize to other materials systems?
- Are there any plans to release energy, stability, or surface property annotations that would allow physically grounded evaluation beyond geometry?
- What are the key insights for future model design drawn from the observed failures—what inductive biases or architectural features are missing?

---

### Official Review · Reviewer_iFXq · 2025-11-01

**Soundness:** 1
**Presentation:** 2
**Contribution:** 2
**Rating:** 2
**Confidence:** 4

**Summary:**

This work introduces a benchmark for evaluating generative models on a perovskite hydride nanoparticle dataset with controlled size and orientation splits. It assesses several generative models for crystal structures using structure-aware metrics for two tasks: (i) generating nanoparticles from a given unit cell and particle radius, and (ii) recovering lattice parameters and space group from a provided structure.

**Strengths:**

- Well-motivated effort to create a nanoparticle dataset and a large, systematically constructed collection.
- Public code and data will be valuable for the materials modeling community.

**Weaknesses:**

My current score is limited by concerns about task definitions and benchmark design; I will adjust the score once these issues are addressed.

- The motivation in Sections 1 and 2.1 is clear, but the central scientific challenges for nanoparticles, such as surface distortions and reconstructions from under-coordinated atoms, are not captured in the benchmark. Although the Limitation section acknowledges this "ideality," the absence of surface reconstructions weakens the scientific relevance and undermines the claim that the benchmark reflects realistic generative modeling needs. Specifically:
    - Task 1 is almost deterministic. Nanoparticles are generated by carving a sphere from a periodic supercell. A simple rule-based algorithm could reproduce the targets without learning a distribution. Without surface non-idealities, the task does not meaningfully require a generative model.
    - Task 2 is an inverse problem, essentially supervised regression + classification (lattice and space group prediction), or a constrained optimization with symmetry detection. It is not inherently generative.
    - If the benchmark is meant to justify generative approaches, it should either (i) include realistic, multimodal nanoparticle physics where sampling uncertainty matters, or (ii) redefine the tasks so that models receive far less structural information (e.g., only composition).
- Because the tasks are not naturally generative, it is unclear why prior generative models are used as baselines. After inspecting the provided code, I found that the implementations are substantially modified "reinterpretations" of the cited methods rather than faithful reproductions. For example, DiffCSP and FlowMM were originally based on EGNN, and MatterGen on GemNet, but the benchmark implementations replace these with different architectures and do not use the official libraries. Since the reported results reuse the original model names, the deviations should be clearly explained in the main text, not only in the code.

**Questions:**

- The carved nanoparticles can become off-stoichiometric under the current procedure. Could this affect the generative task or the physical validity of the outputs?
- In the lattice parameter task, the model predicts three lengths and three angles, yet the reported RMSE is 32–64 Å, far larger than the typical lattice constant range (3–7 Å, up to ~17 Å). This suggests a possible unit/scale mismatch or an implementation issue in the RMSE calculation. In addition, only cubic and orthorhombic cells are included, so all angles are 90°, making angle prediction effectively trivial.
- For joint recovery, exact matching of real-valued properties is required. Please clarify the numerical tolerance used to decide correctness.

Minor typos/comments
- p. 3 Fig 1, blue <-> red; lattice parameter is only reported for $a$ and the max unit cell volume (706.5) seems inconsistent with the max $abc$ product
- p. 6 Sec 3.7, a particle $\mathcal{P} \in \mathbb{R}^3$ -> $\mathcal{P} \in \mathbb{R}^{n \times 3}$

**Details Of Ethics Concerns:**

The code repository linked in the abstract contains a license file that reveals the author’s identity, which violates the anonymity policy.

---

### Official Review · Reviewer_HVMT · 2025-11-01

**Soundness:** 2
**Presentation:** 3
**Contribution:** 2
**Rating:** 2
**Confidence:** 3

**Summary:**

This paper introduces PEROV-H3, a benchmark designed to evaluate generative models' generalization performance between crystalline unit cells and finite nanoparticles of ABH3 perovskites. The authors construct a dataset of 100 DFT-relaxed unit cells and over 210,000 nanoparticles spanning controlled radii (6–30 Å) and orientations, defining two bidirectional tasks—unit-cell → nanoparticle generation and nanoparticle → lattice reconstruction—with structure-aware metrics for geometry and symmetry. Experiments with state-of-the-art models (CDVAE, DiffCSP, FlowMM, MatterGen-MP, ADiT) show that while some achieve low structural errors on intermediate sizes, all models degrade significantly on out-of-distribution radii, with severe failures in symmetry and lattice recovery. These results demonstrate that PEROV-H3 effectively exposes current models’ limitations in capturing the physics of scale and symmetry in generative crystal modeling.

**Strengths:**

1. The authors present well-designed, physics-aware splits and metrics (size extremes, rotation disjointness; structural metrics for both local/global geometry) for the benchmark.

2. Bidirectional tasks that jointly test surface distortions and symmetry/lattice recovery—rare in existing benchmarks.

**Weaknesses:**

1. The experimental results, especially Table 1, for the crystal structure prediction task are suspicious. In both the in-distribution and out-of-distribution settings, CDVAE achieves 10^3  times better performance than the other models (including more advanced ones such as FlowMM and MatterGen). The evaluation metrics of the rest of the models are uniformly poor and nearly identical, e.g., the performance of FlowMM and DiffCSP are almost exactly the same. The authors do not provide a clear or convincing explanation for why these baseline models perform so (equally) poor in this scenario.

2. The structures in the proposed benchmark are only in the form of ABH3, which limits the applicability of the benchmark.

**Questions:**

Can the authors explain the experimental results mentioned in Table 1?

---

### Official Review · Reviewer_JSzy · 2025-11-01

**Soundness:** 2
**Presentation:** 3
**Contribution:** 2
**Rating:** 2
**Confidence:** 2

**Summary:**

The paper introduces PEROV-H3, a benchmark for materials generative models. The benchmark consists of 100 different perovskite materials that were used to construct nanoparticles. Two tasks were defined, one to predict the nanoparticle from the unit cell and one to predict the unit cell from the nanoparticle (lattice parameters and symmetry group). Multiple evaluation metrics were defined for each task, and a comprehensive set of baseline models was tested.

**Strengths:**

- Developing new benchmarks for generative models in materials science is an important research area.
- The paper is well written and thorough on the ensuring quality ID and OOD splits.
- The paper evaluates a comprehensive suite of baseline models.

**Weaknesses:**

- Although this work addresses the important need for new benchmarks in generative materials modeling, the correlation between the proposed tasks and practical utility has yet to be established. It would greatly strengthen the paper if the authors were able to demonstrate the utility of the tasks in practical materials science applications such as hydrogen storage.
- The overall diversity of the benchmark is limited. Perovskites represent a small subset of overall diversity in materials.

**Questions:**

1. Did you reimplement all the baseline models? Or use existing implementations?
2. How do the tasks described in the paper contribute to understanding hydrogen storage in Perovskites? Additionally, how would a model excelling at these tasks accelerate Perovskite-based hydrogen storage research?

---

### Note · Authors · 2025-11-12

I have read and agree with the venue's withdrawal policy on behalf of myself and my co-authors.